# COVID-19 Preparedness and Perceived Safety in Nursing Homes in Southern Portugal: A Cross-Sectional Survey-Based Study in the Initial Phases of the Pandemic

**DOI:** 10.3390/ijerph18157983

**Published:** 2021-07-28

**Authors:** Óscar Brito Fernandes, Pedro Lobo Julião, Niek Klazinga, Dionne Kringos, Nuno Marques

**Affiliations:** 1Department of Health Economics, Corvinus University of Budapest, Fővám tér 8, H-1093 Budapest, Hungary; 2Department of Public and Occupational Health, Amsterdam Public Health Research Institute, Amsterdam UMC, University of Amsterdam, Meibergdreef 9, 1105 AZ Amsterdam, The Netherlands; n.s.klazinga@amsterdamumc.nl (N.K.); d.s.kringos@amsterdamumc.nl (D.K.); 3Public Health Research Centre, NOVA National School of Public Health, Universidade NOVA de Lisboa, Avenida Padre Cruz, 1600-560 Lisboa, Portugal; 4Algarve Biomedical Center, Campus Gambelas, University of Algarve, 8005-139 Faro, Portugal; pmjuliao@ualg.pt (P.L.J.); nsmarques@ualg.pt (N.M.); 5Faculty of Medicine and Biomedical Sciences, Campus Gambelas, University of Algarve, 8005-139 Faro, Portugal

**Keywords:** SARS-CoV-2, care home, long-term care, social care, public health emergency, preparedness, contingency plan, safety culture, workforce, survey

## Abstract

(1) Background: Nursing homes’ preparedness in managing a public health emergency has been poor, with effects on safety culture. The objective of this study was to assess nursing homes’ COVID-19 preparedness in southern Portugal, including staff’s work experiences during the pandemic. (2) Methods: We used a COVID-19 preparedness checklist to be completed by management teams, followed by follow-up calls to nursing homes. Thereafter, a survey of staff was applied. Data analysis included descriptive statistics, exploratory factor analysis, and thematic analysis of open-end questions. (3) Results: In total, 71% (138/195) of eligible nursing homes returned the preparedness checklist. We conducted 83 follow-up calls and received 720 replies to the staff survey. On average, 25% of nursing homes did not have an adequate decision-making structure to respond to the pandemic. Outbreak capacity and training were areas for improvement among nursing homes’ contingency plans. We identified teamwork as an area of strength for safety culture, whereas compliance with procedures and nonpunitive response to mistakes need improvement. (4) Conclusions: To strengthen how nursing homes cope with upcoming phases of the COVID-19 pandemic or future public health emergencies, nursing homes’ preparedness and safety culture should be fostered and closely monitored.

## 1. Introduction

It has been more than a year since the World Health Organization (WHO) declared the coronavirus outbreak a pandemic. Currently, as the pandemic unfolds and national vaccination campaigns intensify, countries are still struggling to contain the spread of different virus strains and their effects on the population’s health. Each country has been facing its specific challenges, depicted by the large variability in countries’ set of responses to this global health crisis [1,2,3]. A common denominator across countries is how older people are vulnerable to this pandemic. In many countries, nursing homes are of special concern given their residential setting and the characteristics of their users [4,5]. Thus, nursing homes are particularly prone to becoming an epicenter for an outbreak of coronavirus [3,6].

The effects of the pandemic in nursing homes worldwide have taken a toll on all of us and in our humanity, as these facilities have been affected in an unprecedented manner [5]. The initial responses to the coronavirus disease (COVID-19) in nursing homes were limited. This depicts a poor preparedness of nursing homes in managing a public health crisis, despite past serious concerns of potential threats [7,8] with negative effects on the safety of staff and residents, exposing them to a hazardous and potentially life-threatening environment on a daily basis. This concerning unpreparedness uncovered long-lasting caveats in nursing homes, such as insufficient coordination with the health care system, generalized underfunding, and insufficient and inadequate workforce in the sector [9,10].

Whilst caring for residents, nursing home staff are also managing concerns about their health and well-being and the safety of their family members. In the first stages of the pandemic, the focus on the health and well-being of nursing home staff was not immediately prioritized. Only later, balancing safety, well-being, and quality of life among staff became a topic of discussion, leveraging from the focus attributed to these topics by international organizations, such as the Organization for Economic Co-operation and Development (OECD) [9] and WHO [11]. To mark the World Patient Safety Day on 17 September 2020, the latter even chose the theme ‘Health Worker Safety: A Priority for Patient Safety’. This theme is aligned with evidence suggesting that a stronger perception of safety culture (i.e., the product of individual and group values, attitudes, perceptions, competencies, and patterns of behavior that shape safety management) is associated with greater care quality to nursing home residents [12]. In addition, it is reasonable to expect that safety culture has a large influence on staff job satisfaction, turnaround, and well-being. However, little is known about safety culture in long-term care facilities [13,14,15].

In Portugal, the first death from COVID-19 occurred on 16 March 2020. During the initial phases of the pandemic, 40% (450/1126) of the deaths from COVID-19 were among residents in nursing homes [16]. At that time, the testing of residents and staff for COVID-19 was slow, and coordination between agencies was reportedly poor. By mid-March 2020, the Algarve Biomedical Center (ABC) was commissioned by the Ministry of Labor, Solidarity, and Social Security (MLSSS) to test staff and residents at nursing homes in southern Portugal, whether these facilities were operating under a valid license or not. In total, 194 nursing homes from southern Portugal (Algarve and Alentejo regions) benefited from this protocol.

The main goal of this study is to assess the COVID-19 preparedness of nursing homes in southern Portugal and explore its effects on nursing home staff safety culture and well-being in the early phases of the pandemic (from March to July 2020). Thus, the objectives of this study are: (1) to assess COVID-19 preparedness of nursing homes in two regions of southern Portugal (Algarve and Alentejo) in the early phases of the pandemic; (2) to better understand safety concerns and well-being of nursing home staff; and (3) to understand nursing home staff work experiences during the pandemic, including resident safety culture. Findings will be used to inform current ongoing responses targeting nursing homes, optimize dealing with upcoming phases of the SARS-CoV-2 pandemic, and strengthen nursing home preparedness to other public health emergencies.

## 2. Materials and Methods

### 2.1. Study Design and Population

We conducted a cross-sectional survey-based study in nursing homes of 2 regions in southern Portugal (Algarve and Alentejo) in the early phases of the COVID-19 pandemic (from March to July 2020). Nursing homes were identified by the regional offices of the Social Security Institute. Nursing homes participated in our study voluntarily and independently of parallel ongoing COVID-19 testing initiatives. Participating nursing homes self-assessed COVID-19 preparedness using a checklist filled by managers; thereafter, a web-based self-administered survey was conducted among staff. For this observational study, a formal ethics approval was waived because of the non-intrusive and non-interventional characteristics of the study; the anonymization of all personal data prior to analysis and reporting; the existing protocol between the Social Security Institute and the ABC research institute; and the urgency of gathering key information about the problems that nursing homes had been facing in a timely fashion. Nursing home staff responding to our survey granted consent at two moments: before answering the survey and when submitting their answers.

### 2.2. Instruments

#### 2.2.1. COVID-19 Preparedness Checklist

The checklist (Appendix A) was developed based on that of the Centers for Disease Control and Prevention (USA) [17]. Cultural and contextual adaptations were considered during translation (e.g., we included or adapted items in the checklist to reflect the guidelines and orientations issued by the Portuguese Directorate—General of Health). The checklist encompassed four parts: (1) nursing home characteristics; (2) structure for planning and decision-making; (3) development of a contingency plan; and (4) general features of the contingency plan. In April 2020, the checklist was sent to nursing homes. A 2-stage voluntary-basis engagement followed: first, nursing home managers self-assessed the COVID-19 preparedness of their respective facilities by using the checklist and submitted their responses to the research team; second, a research team representative set up a follow-up video/phone call with nursing home managers for a checklist walkthrough discussion (Appendix A). At this time, Portugal was experiencing its third emergency state period; the first was declared on 18 March 2020, and people were instructed to remain at home with few exceptions; the third emergency state ended on 3 May. By the end of that period, Portugal reported 25,524 cases and 1,063 deaths by COVID-19, of which 87% were among people aged 70 years or more (lethality rate of 4.2% for the general population vs. 14.9% for people aged 70+) [18]. Visiting was not allowed in nursing homes (from 16 March to 17 May 2020), 1757 residents and 141 workers had been infected with SARS-CoV-2, and the death tolls were 243 in nursing homes [18].

#### 2.2.2. Staff Survey

The *Safety concerns and well-being of nursing home personnel* survey (Appendix A) was developed by the research team for the purpose of this study. Various sections of the survey drew on existing validated instruments in the Portuguese language, such as the nursing home survey on patient safety culture developed by the Agency for Healthcare Research and Quality (USA) [19], the WHO-5 Well-Being Index [20], and the Minimum European Health Module [21]. Novel questions and other survey design features (e.g., question ordering) were previously discussed with experienced international researchers in long-term care. We asked respondents through an 11-point Likert scale (from 0 = Very low to 10 = Very high) how they perceived the risk of becoming infected with coronavirus and becoming severely ill in case of infection. A 10-item list with a 6-point Likert scale (from 1 = All of the time to 6 = At no time) assessed the extent to which staff worked in an environment where fear/anxiety and unusual absenteeism among peers were present. Two items were positively worded (item 4 and 6), for which we used reversed coding. Lower scores across items suggest greater perceived fear/anxiety and absenteeism amongst staff. We used an 18-item set from the nursing home patient safety culture questionnaire with a 5-point Likert scale (from 1 = Strongly disagree to 5 = Strongly agree) and a “Does not apply/Don’t know” answer option. Items were grouped into 5 safety culture composites: teamwork, staffing, compliance with procedures, training and skills, and nonpunitive response to mistakes [19]. The WHO-5 Well-Being Index was computed as the composite raw score that resulted of totaling the scores of the answers to 5 statements using a 6-point Likert scale (from 0 = At no time to 5 = All of the time) on how a respondent had been feeling over the last two weeks; a score below 13 was indicative of poor well-being. The Minimum European Health Module asked about respondents’ self-perceived health, chronic morbidity, and activity limitations. We also asked about respondents’ characteristics, those of their job and household, and in which areas there was a need for support from others. The last question in the survey was an open-ended question where respondents could share, in their own words, how they have been experiencing the pandemic crisis.

We conducted pre-testing and cognitive testing, involving 1 interviewer and 3 respondents representing essential care workers. Survey data collection started after the completion of an initial round of COVID-19 testing across nursing homes: from 19 May to 9 June 2020 in Algarve and from 29 June to 21 July 2020 in Alentejo. A weblink to the survey was sent to all nursing homes; thereafter, the link was disseminated among staff through internal communication channels. Follow-up calls and emails occurred one week prior to the data collection due date. On the last day of administering the survey, 389 cases had been reported in Algarve (11‰ of total confirmed cases in Portugal) [22], which represented a 9% increase from the first day of the survey [23]; in Alentejo, 636 cases had been reported by the last day of administrating the survey (13‰ of total confirmed cases) [24], which represented a 33% increase from the first day of the survey [25].

### 2.3. Statistical Analysis

We used descriptive statistics to characterize attributes of participating nursing homes and survey respondents and to examine missing data patterns. We summarized the share of facilities that had fully implemented each set of items in the preparedness checklist by computing the geometric mean across items. Composite response frequencies of the resident safety culture from the perspective of staff were computed by averaging positive responses. We used a spreadsheet to group data from open-ended questions from participating nursing homes and survey respondents into meaningful emerging categories.

We used Goodman and Kruskal’s Gamma test to measure the correlation between ordinal variables in the survey. We performed exploratory principal axis factor analysis via oblimin rotation with Kaiser normalization on the 10-item questions about perceived fear/anxiety and absenteeism amongst staff. Missing values were excluded listwise. The number of factors was determined based on parallel analysis, Kaiser criterion of eigenvalues (eigenvalues greater than 1 were kept), and a visual inspection of the scree plot. We assessed the final factor loading structure for internal consistency with Cronbach’s alpha. Thereafter, we conducted a K-means cluster analysis with the regression method.

We used IMB SPSS Statistics version 26 (IBM Corp., Armonk, NY, USA) for all analyses. The confidence level was set at 95%.

## 3. Results

### 3.1. Nursing Home COVID-19 Preparedness

Between March and July 2020, 96 nursing homes in Algarve (all the 81 licensed and 14 non-licensed facilities) and 99 licensed nursing homes in Alentejo were engaged in the COVID-19 initiatives set forth by the ABC (Figure 1). A total of 71% (*n* = 138) of nursing homes returned the COVID-19 preparedness checklist: return rate was of 53% in Algarve (51/96) and 88% in Alentejo (87/99). In parallel, we conducted 83 follow-up calls with nursing homes, of which 65 (78%) were with a facility that had previously returned its COVID-19 preparedness checklist. In total, 720 nursing home staff replied to the safety and well-being survey.

On average, a quarter of nursing homes did not fully observe an adequate structure for planning and decision-making in response to the pandemic, and 17% did not have a fully implemented and disseminated contingency plan, with designated key people to operationalize thereof (Table 1). The key areas of a contingency plan oftentimes overlooked were those of outbreak capacity and education and training. On average, 41% of nursing homes fully fulfilled the items grouped under ‘outbreak capacity’, often overlooking postmortem care and morgue capacity planning. With regards to education and training, 43% of nursing homes had a training plan implemented to address the needs of key people (residents, staff, volunteers, family members, or visitors). A full breakdown of the items in the checklist is available in Appendix A.

Based on the returned checklists and follow-up calls with nursing homes, we identified key items that were most frequently overlooked:poor communication channels, both internal and external, often failing to disseminate the contingency plan among key stakeholders (e.g., staff) and engaging with health and other competent authorities;inexistent or poor planning to isolate or transfer residents if need be;poor surveillance systems to monitor for symptoms among residents and staff;insufficient planning to overcome hindrances related to staff shortages and absenteeism, and infrastructure constraints (e.g., bed overcapacity in isolation rooms);the inexistent monitoring system of the effectiveness of the measures aiming at addressing behavioral factors, both at the institutional and individual level;misuse of personal protective equipment (PPE) attributed to poor training and a generalized shortage of specific equipment (e.g., gowns and FFP2 face masks).

We also highlight good practices found at some facilities:
continuous revision of the contingency plan to reflect any updates to the guidelines set forth by the Directorate—General of Health and other relevant competent authorities;emergency protocol with the nearest primary health care centers for a quick response in case of an outbreak;systematically maintaining an inventory of PPE in close collaboration with governmental authorities;using social media and other platforms to update families and carers on residents’ well-being and on the public health measures that the nursing home is developing.

A detailed list is available in Appendix A.

### 3.2. Perceived Safety and Well-Being among Nursing Home Staff

Most of the 720 survey respondents were female (93%), and the average age was 45 years old (Table 2). The highest educational level attained by 41% (297/720) of respondents was lower than secondary education, and 29% (212/720) concluded university studies. Most of the respondents (606/720, 84%) self-reported good or fair health; only ten people (1%) referred to their health as bad or very bad. The average well-being index was 15 (out of 25), and 22% (158/720) of the respondents reported having a longstanding health problem. Respondents’ household context varied: 20% (147/720) lived with people older than 65 years old, 35% (255/720) lived with children less than 12 years old, and 22% (161/720) lived with people considered to be essential workers. Workwise, most of the respondents worked directly with nursing home residents (525/720, 73%), had a long-lasting work contract (399/720, 55%), and worked full-time (649/720, 90%).

Our data suggested a strong positive association (G = 0.530; *p* < 0.001) between the risk perception of becoming infected with coronavirus and that of becoming severely ill with COVID-19 (Figure 2). Nursing home staff perceived that the COVID-19 testing of both residents and workers was a suitable approach towards supporting nursing homes (G = 0.884; *p* < 0.001). Our results did not suggest a strong or significant association between the importance of being tested for COVID-19 and the respondents’ risk perception of becoming infected or severely ill with COVID-19.

### 3.3. Fear and Absenteeism Attributed to COVID-19

Our data suggested a correlation of 0.336 between fear and absenteeism attributed to COVID-19 (Appendix A). We observed six clusters for respondents’ perception of fear and absenteeism (Figure 3). A total of 170 respondents were in cluster 1, 17 in cluster 2, 123 in cluster 3, 68 in cluster 4, 188 in cluster 5, and 33 in cluster 6. Respondents in clusters 1, 3, and 5 were homogeneous with regards to their perception of absenteeism amongst staff, yet fear perception varied widely. On the other hand, respondents in clusters 4 and 6 perceived fear amongst staff in a comparable manner (i.e., they perceived colleagues experiencing fear most of the time), yet respondents in cluster 6 perceived greater absenteeism among colleagues relative to respondents in cluster 4. Respondents in clusters 2 and 4 were younger than average, with the former being the youngest group (on average). Respondents from clusters 1 and 2 showed above-average well-being, whereas respondents in cluster 6 showed on average the lowest well-being level. The remaining clusters showed similar close to zero Z-scores regarding age and well-being.

### 3.4. Experiences with Coping with the Pandemic

We grouped into three categories the 71 answers to the open-end survey question on how respondents were experiencing the pandemic. The first category focused on how respondents perceived support from competent authorities to inform decision-making. Some respondents signaled the poor coordination between public health authorities (Ministry of Health) and regional offices of the Social Security Institute (MLSSS), which resulted in increased administrative burden and increased waiting time in receiving support. Some respondents also perceived that the competent authorities had little knowledge of the real context and circumstances of most nursing homes, and thus, were not aware of the structural hindrances these institutions were facing to follow many of the guidelines and orientations set forth by health authorities.

A second category focused on the psychological support available to residents and staff throughout the pandemic crisis, or the lack thereof. The availability of psychological support to staff was highlighted as an important tool for managing stress levels and the emotions of colleagues, residents, and those of residents’ families and carers. Some respondents also highlighted increased levels of stress and anxiety amongst nursing home managers, which made them more reactive and less supportive towards staff. Notwithstanding, some respondents found in other colleagues the much-needed support to cope with fear and other daily challenges.

The third category refers to a generalized perception of misinformation in regard to the contingency plan, where respondents identified not knowing its content or how to best proceed to activate and comply with the foreseen planning. Related to this is the common misuse of PPE, which could signal shortages in PPE, as well as inadequate and insufficient training about its appropriate use.

We observed statistically significant differences in the proportion of respondents from Algarve and Alentejo identifying the need for psychological support (Algarve: 1:67 vs. Alentejo: 1:25; *p* = 0.035) and support with dealing with fear/anxiety (Algarve: 1:4 vs. Alentejo: 1:3; *p* = 0.014) (Appendix A).

### 3.5. Nursing Home Resident Safety Culture

We computed composite percent positive scores related to the constructs of teamwork, staffing, compliance with procedures, training and skills, and nonpunitive response to mistakes (Figure 4). The lowest composite percent positive was relative to compliance with procedures (51%), followed by nonpunitive response to mistakes (55%) and staffing (60%). The largest composite percent positive was that of teamwork (78%), followed by training and skills (70%). A detailed scoring list of safety culture survey items and respective percent positive, neutral, and negative composites for Algarve and Alentejo is available in Appendix A.

## 4. Discussion

We examined nursing home COVID-19 preparedness and the staff’s perceived well-being and safety, including nursing home resident safety culture, in the early phases of the pandemic, in two regions of southern Portugal. Our findings suggest that COVID-19 preparedness in nursing homes was poor. Often, nursing homes overlooked the importance of planning a thorough contingency plan and of having an adequate structure for planning and decision-making to respond to the COVID-19 pandemic. In general, nursing home staff showed concerning well-being scores. Our data support that staff perceived fear and absenteeism among peers differently, the latter with greater variability. We identified teamwork as a strength for resident safety culture, whereas compliance with procedures, nonpunitive response to mistakes, training and skills, and staffing were signaled as areas for improvement.

Some nursing homes did not use the preparedness checklist as a tool to inform the preparation of a contingency plan. We explain non-response by considering three key factors. First, the use of the preparedness checklist by nursing homes occurred on a voluntary basis and was independent of getting access to COVID-19 testing. Second, nursing homes had to quickly react to threats posed by COVID-19 and adapt their operations; hence, non-responders may have felt overconfident with the actions implemented in their facility and did not assign substantial importance to other initiatives wherein no additional value is perceived. And third, by the time when news in the Portuguese media started linking fragilities in nursing homes’ contingency plans and COVID-19 outbreaks, these facilities were overwhelmed with daily mandatory and uncoordinated data requests from different agencies; contrary to governmental data-driven initiatives targeting hospitals, this was not the case for social care. Contributing to this context could be the current levels of integration of health and social care information, which is uncoordinated and poor regarding data interoperability [26]. In the future, initiatives for collecting actionable performance intelligence [27] in a timely fashion should be outlined in a concerted manner, prioritizing those data that could inform context-specific decision-making [28], namely via actionable dashboards [29].

Our findings suggest that many nursing homes overlooked the importance of thorough and exhaustive planning and the role of the contingency plan in that. For example, in Portugal, one of the most serious events in a nursing home which resulted in the death of 18 people seems to have been strongly related to the absence of a proper contingency plan; a few months later, little had improved [30]. This is concerning when our data suggest that many other nursing homes did not have a suitable decision-making structure to cope with the pandemic. Furthermore, planning for key aspects of a contingency plan was overlooked, such as those of outbreak capacity, education and training, proper in-house communication mechanisms with staff, and adequate use and access to PPE. These aspects contributing to the overall unpreparedness of nursing homes were of concern also internationally [6,11,31,32,33,34], particularly for their effect on the commitment of nursing homes in keeping residents safe, socially active, and provide positive care experiences. To address these concerns, and informed by preliminary results of this study, the MLSSS in October 2020 commissioned a national support line operating 24/7 [35], and in December 2020, a national training program on outbreak management targeted at social workers in nursing homes [36]. The training program is expected to strengthen adherence to and compliance with safety measures in nursing homes (e.g., using PPE properly), minimizing risks and strengthening resident safety culture from the staff’s viewpoint. These initiatives highlight the potential of partnerships with academia (e.g., research centers such as the ABC), which may have contributed to control the deaths among nursing home residents. In Portugal, since the onset of the pandemic until 4 February 2021, 28% (3750/13,482) of deaths were among nursing home residents [37], which compares well relative to other countries that were far more affected in that population group [16].

Many of the vulnerabilities in the COVID-19 preparedness of nursing homes occur atop of long-lasting structural barriers (e.g., overcrowding and staff shortages) [26,38], which could exacerbate the effects of the pandemic in nursing homes. Although the MLSSS funded a program (MAREESS) for the emergency recruitment of staff to social facilities [39], the context of decades of disinvestment, not only financially but also strategically, is difficult to address. The scarcity of policies addressing the training and retention of people with adequate skill mix to manage and work in these facilities depict a lack of a shared strategic vision on how nursing homes fit in the social and health care systems [10]. There is an urgency in effectively bridging social and health care, nudging nursing homes towards more integrated care pathways. This transition could strengthen safety, support, and care quality for the elderly, while nursing homes could become more attractive workplaces.

Nursing home staff perceived their working realities on fear and absenteeism very differently. Several features may influence how staff are affected by psychological distress (e.g., staff-to-bed ratio, access to PPE, safety guidelines, and professional support at the workplace [40,41,42]) and its effects on absenteeism. The extent to which facility and individual-level characteristics contribute to this variability amongst staff remain. Hence, one-size-fits-all actions to mitigate effects attributed to fear and absenteeism may have little or no consequences in psychologically hazardous environments; rather, a dynamic approach to bolster staff in accessing their resilience should be set forward [43,44]. The design of such a set of actions also should be abreast of greater communication mechanisms, including those with competent and governing authorities, but also internally. Strengthening communication channels with staff, seeking to understand their priorities, experiences with coping with the pandemic, and how to best support them could function as a *psychological PPE* [45], nurturing, and instilling hope among staff. This was crucial at the early stages of the pandemic and will remain key throughout.

Safety culture studies in social care settings such as nursing homes are widely lacking, and much of the available literature refers to the USA context [14]. To the best of our knowledge, there is no peer-reviewed study focusing on safety culture in Portuguese nursing homes. Data availability on the safety culture of nursing homes could prove a crucial step towards informed decision-making by signaling strengths and areas for resident safety culture, which may also reflect on staff’s safety, well-being, job satisfaction, and turnover [46]. Our findings highlight several areas for safety culture improvement, such as compliance with procedures, nonpunitive response to mistakes, and staffing. In our sample, the composite average positive percent for compliance with procedures (51%) and nonpunitive response to mistakes (55%) were lower than those for nursing homes in Belgium (50% and 61%) [47] and Norway (62% and 71%) [48]. Conversely, the composite average positive percent for staff was greater in our sample (60%) relative to those nursing homes in Belgian (38%) and Norwegian (44%) studies. In addition, top average positive composites in our sample—teamwork (78%) and training and skills (70%)—were greater than those reported in [47] (76% and 65%) and [48] (68% and 51%). These indicators could be used to benchmark and steer where nursing home improvement initiatives are needed the most. The effects of such improvements could have major implications, namely in staff turnover and residents’ care quality [49]. However, we suspect that systemic and long-lasting effects in nursing home resident safety culture in Portugal could be achieved not only with greater investment but also with a change of the societal opinions and the public perception of nursing homes.

### Study Strengths and Limitations

Our study strength relies on the large sample of nursing homes involved, the engagement of staff with our survey, and the use of safety culture indicators. Also, we featured in our survey international standardized measures in hopes of future possible international comparisons. However, this study should be interpreted considering some limitations. First, our sample is not representative of the nursing home population and social and health care workers in these facilities in Portugal, and thus, hampering the generalizability of our findings outside the context of the study. Notwithstanding, these data, in parallel to the COVID-19 testing, were crucial to inform and support health and social authorities in better supporting nursing homes; also, these findings are well aligned with those of other international studies. Second, we only used one section of the resident safety culture questionnaire to reduce the length of our survey. By doing so, we are aware that other areas that may be significantly affecting the safety culture of a nursing home may not have been captured by our data. Finally, we understand that the data collection method chosen for our survey may have limited the ability of some people responding to the survey, particularly people without access to a computer or smartphone and with lower literacy on information and communications technology. Also, we are aware that to increase the response rate, some nursing homes had a computer available for staff interested in answering the survey. The extent to which this commodity influenced their answers was not considered in data analysis.

## 5. Conclusions

The objective of this study was to assess the COVID-19 preparedness of nursing homes in southern Portugal in the early phases of the pandemic, including the work experiences and safety concerns of staff. In Portugal, COVID-19 seems to have exacerbated the longstanding and systemic fragility of social care and the underinvestment in high-quality long-term care, including nursing homes. That disinvestment is partly exposed by the generalized unpreparedness across nursing homes and their difficulties with keeping residents and staff safe during the pandemic. Hence, rethinking the positioning of nursing homes in the social and health care systems could potentially strengthen nursing home preparedness. In addition, future response actions to the pandemic should involve nursing home representatives in decision-making processes to the extent where it is feasible and leverage from available evidence at the national and international level.

Communication between competent authorities and nursing homes should be clearer in scope, actions, time, and responsibilities, increasing transparency and accountability in responding to the challenges of the COVID-19 pandemic. Improved communication channels are also key inside nursing homes to better discuss the effects of the pandemic within the facility and as a mechanism to bolster staff resilience. Additionally, assessing safety culture amongst nursing home staff can be a meaningful approach to understanding and dealing with staff’s experiences of fear and anxiety. These could be addressed by involving staff in decision-making, recognizing their efforts and merits, enhancing collective learning and sharing, learning the current priorities of staff, and normalizing the sharing of feelings of fear, anxiety, and psychological frailty. Dealing with a sudden threat like the first phase of the COVID-19 pandemic posed challenges to the nursing home community as a whole, involving residents, relatives, social and health care workers, and management alike. Lessons learned as discussed in this study on preparedness and perceived safety should result in a more resilient nursing home sector in Portugal for the challenges still to come.

## Figures and Tables

**Figure 1 ijerph-18-07983-f001:**
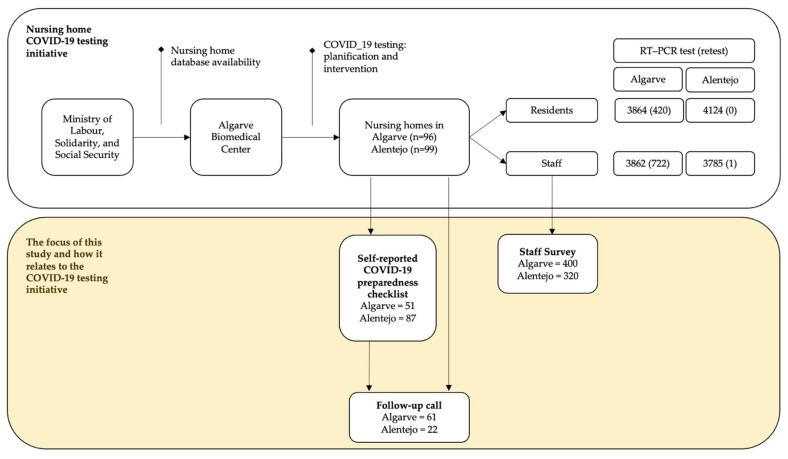
Descriptive frequencies of the nursing home COVID-19 testing initiative and the unfolding of this study, in Algarve and Alentejo (Portugal), amid the initial phases of the pandemic (March to July 2020).

**Figure 2 ijerph-18-07983-f002:**
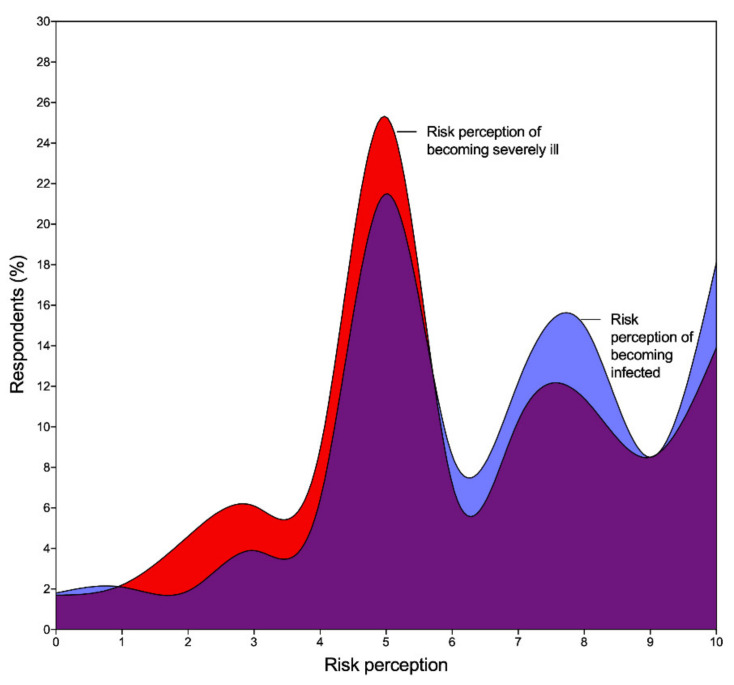
Distribution of the overall risk perception of becoming infected with coronavirus and becoming severely ill with COVID-19 and respective overlapping.

**Figure 3 ijerph-18-07983-f003:**
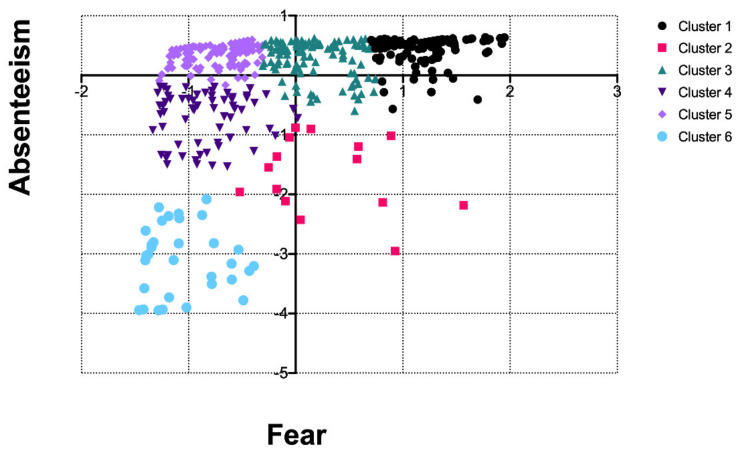
Clusters based on fear and absenteeism attributed to COVID-19 (Z-scores).

**Figure 4 ijerph-18-07983-f004:**
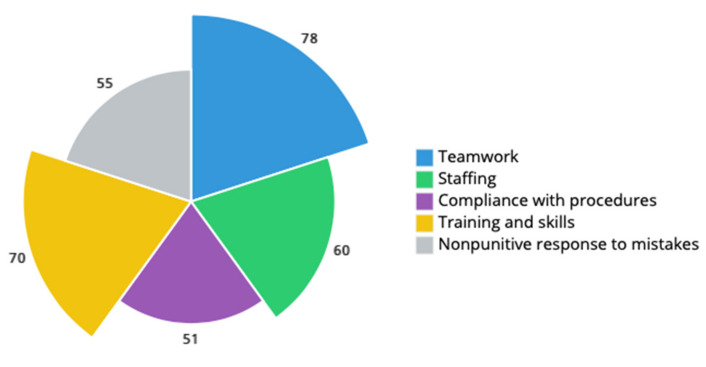
Nursing home resident safety culture composite average positive response (%).

**Table 1 ijerph-18-07983-t001:** Nursing home COVID-19 preparedness checklist compliance scores.

Item Grouping	COVID-19 Preparedness Compliance (%) ^a^		
	Algarve (*n* = 51)	Alentejo (*n* = 87)	Total (*n* = 138)
Structure for planning and decision-making	65%	79%	74%
COVID-19 contingency plan	75%	87%	83%
**Elements of a COVID-19 Contingency Plan**			
General	66%	72%	70%
Outbreak capacity	35%	45%	41%
Communication	79%	76%	77%
Supplies and resources	68%	79%	75%
Education and training	44%	43%	43%
Occupational health	71%	75%	74%
Identification and management of ill residents	87%	81%	83%
Access control	83%	81%	82%

^a^ Scores were computed as the geometric mean of items fully implemented within each group.

**Table 2 ijerph-18-07983-t002:** Characteristics of the respondents to the safety concerns and well-being of nursing home staff survey, and those of their household and work contexts.

		Total		Missing Data ^a^	
		*N* = 720			
		*n*	%	*n*	%
**Individual**	Sex				
	Female	667	93	7	1
	Male	46	6		
	Age				
	Mean (SD ^b^)	45 (11)		3	0.4
	Education ^c^				
	Primary	297	41	12	1.7
	Secondary	199	28		
	Tertiary	212	29		
	Self-perceived health status				
	Very good	104	14	0	0
	Good	365	51		
	Fair	241	33		
	Bad or Very bad	10	1		
	WHO Well-Being Index				
	Mean (SD)	15.5 (5.5)		0	0
	Longstanding health problem				
	Yes	158	22	72	10
**Household**	Living with people aged 65 and over				
	Yes	147	20	8	1.1
	Living with children (up to 12 years old)				
	Yes	255	35	2	0.3
	Living with people in a professional group with increased risk				
	Yes	161	22	5	0.7
**Work**	Works directly with residents				
	Yes	525	73	0	0
	Work contract duration				
	Less than a year	100	14	0	0
	1 to 2 years	94	13		
	3 to 5 years	127	18		
	More than 5 years	399	55		
	Weekly working hours				
	Less than 20 h	52	7	0	0
	21 h up to 31 h	19	3		
	More than 31 h	649	90		

^a^ Missing data also include answer options ‘Decline to answer’ or ‘I don’t know’. ^b^ SD: Standard deviation. ^c^ Primary education: up to 9 years of formal education; secondary education: 12 years of education; tertiary education: university education.

## Data Availability

The data presented in this study are available on request from the corresponding author.

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
