# Peer review of "COVID-19 Preparedness and Perceived Safety in Nursing Homes in Southern Portugal: A Cross-Sectional Survey-Based Study in the Initial Phases of the Pandemic"

_ijerph, 2021, doi:10.3390/ijerph18157983_

Round 1

Reviewer 1 Report

Thank you for your interesting study. Please review the writing once more. In several places the wording is not quite adequate. The use of "fragile" to indicate problematic or downright dangerous situations may need to be reviewed. The term "personnel" could also be replaced with "staff". 

Please explain and define the term "safety culture" in more detail. Thus, the consequences of the problems in safety culture can be more easily identified and understood. 

When describing the staff's situation, it seems relevant to mention the lockdown situations that many countries had to undergo as part of the first response to the pandemic. During this time, the nursing home staff not only had to fulfill their daily work, but also had to take over the social support of residents who were deprived of contacts with their close family members. 

Personnel survey: Was this data collection tool researcher developed or were validated instruments used? It is necessary to clearly indicate this. 

It has been highlighted that nursing homes often lack clear processes of interaction with hospitals or other institutions of the healthcare systems. In the results, there does not appear to be a reference to this. Why were these aspects not studied given the fact that this lack has been commented upon worldwide? 

In the discussion, it is necessary to address the distances of nursing homes to other healthcare organisations such as hospitals. It is equally important to indicate potential ethical issues that have arisen due to the unpreparendness. In addition, please address the problem of education and lack of skills/capacities in the staff and how this influenced the nursing home's response. 

Best wishes. Your reviewer

Author Response

We extend our sincere thanks to the three Reviewers for their constructive suggestions. A point-by-point response to Reviewer 1 follows. All necessary changes have been incorporated into the revised version of the manuscript.

Kind regards,

The Authors

Point-by-point response to Reviewer 1

Thank you for your interesting study. Please review the writing once more. In several places the wording is not quite adequate. The use of "fragile" to indicate problematic or downright dangerous situations may need to be reviewed. The term "personnel" could also be replaced with "staff". 

Response: We revised the use of the term “fragile” throughout and replaced the use of “personnel” with “staff.”

Please explain and define the term "safety culture" in more detail. Thus, the consequences of the problems in safety culture can be more easily identified and understood. 

Response: We now have defined safety culture in the scope of this work. The text now reads as follows:

The latter, on 17 September 2020, to mark the World Patient Safety Day even chose the theme ‘Health Worker Safety: A Priority for Patient Safety.’ This theme is aligned with evidence suggesting that a stronger perception of safety culture (ie, the product of individual and group values, attitudes, perceptions, competencies, and patterns of behavior that shape safety management) is associated with greater care quality to nursing home residents [12]. In addition, it is reasonable to expect that safety culture has a large influence on staff job satisfaction, turnaround, and well-being. However, little is known about safety culture in long-term care facilities [13-15]. (Lines 65 – 72)

When describing the staff's situation, it seems relevant to mention the lockdown situations that many countries had to undergo as part of the first response to the pandemic. During this time, the nursing home staff not only had to fulfill their daily work, but also had to take over the social support of residents who were deprived of contacts with their close family members. 

Response: We thank Reviewer 1 for this suggestion. We have included additional information in-text:

At this time, Portugal was experiencing its third emergency state period; the first was declared on March 18, 2020, and people were instructed to remain at home with few exceptions, and the third emergency state ended on May 3. By the end of that period, Portugal was reporting 25,524 cases and 1,063 deaths by COVID-19, of which 87% were among people aged 70 years old and over (lethality rate of 4.2% for the general population vs 14.9% for people aged 70+) [18]. Visiting was not allowed in nursing homes (from March 16 to May 17, 2020), 1,757 residents and 141 workers had been infected with SARS-CoV-2, and the death tolls were 243 in nursing homes [18]. (Lines 121 – 128)

Personnel survey: Was this data collection tool researcher developed or were validated instruments used? It is necessary to clearly indicate this. (l.123)

Response: We have clarified this information about the survey. The text now reads as follows:

The Safety concerns and well-being of nursing home personnel survey (Supplementary File 3) was developed by the research team for the purpose of this study. Various sections of the survey drew on existing validated instruments in the Portuguese language, such as the nursing home survey on patient safety culture developed by the Agency for Healthcare Research and Quality (USA) [19], the WHO-5 well-being index [20], and the Minimum European Health Module [21]. (Lines 130 – 135)

It has been highlighted that nursing homes often lack clear processes of interaction with hospitals or other institutions of the healthcare systems. In the results, there does not appear to be a reference to this. Why were these aspects not studied given the fact that this lack has been commented upon worldwide?

Response: When discussing the findings of your study we draw attention to the fact that there is a lack of strategic vision on how nursing homes fit both in the social and health systems, signaling a clear urgency in effectively bridging social and health care. Exploring these aspects further was, however, out of the scope of our study, albeit we recognize the importance of the subject.

In the discussion, it is necessary to address the distances of nursing homes to other healthcare organisations such as hospitals. It is equally important to indicate potential ethical issues that have arisen due to the unpreparendness. In addition, please address the problem of education and lack of skills/capacities in the staff and how this influenced the nursing home's response. 

Response: We followed the suggestion of Reviewer 1 and addressed the ethical issues arisen because of unpreparedness and the impact of poor training in safety culture (page 11). We did not consider, at this time, including information about the distance of nursing homes to nearby health care organizations because we understand that this falls out of scope of this study.

Reviewer 2 Report

Dear,

The research theme is relevant to the health area in times of pandemic, but some adjustments and recommendations must be observed for the publication of the manuscript.

In the title I suggest exclude the number "138" and "survey-based 3 study in the initial phases of the pandemic".

I suggest including in the introduction the research question. 

In the introduction, clarify the importance of the safety of professionals and institutionalized elderly people and their relationship with the pandemic. What can pose security risks?

In the method including the type of reserarch with theoretical referencial. 

In the method, it is necessary to include the approval protocol of the Research Ethics Committee, considering that the research involved data from institutions and participants.

In the conclusions is necessary including the reach of objective The objective of this study was to assess nursing homes’ 22 COVID-19 preparedness in Southern Portugal, including personnel’s work experiences during the 23 pandemic. 

Best, Regards, Evaluator

Author Response

We extend our sincere thanks to the three Reviewers for their constructive suggestions. A point-by-point response to Reviewer 2 follows. All necessary changes have been incorporated into the revised version of the manuscript.

Kind regards,

The Authors

Point-by-point response to Reviewer 2

The research theme is relevant to the health area in times of pandemic, but some adjustments and recommendations must be observed for the publication of the manuscript.

In the title I suggest exclude the number "138" and "survey-based 3 study in the initial phases of the pandemic".

Response: We have removed the number of nursing homes from the title but have decided to keep the subtitle as is. We believe that the subtitle provides key information to the reader about the methods used and when in the pandemic is this study referring to.

In the introduction, clarify the importance of the safety of professionals and institutionalized elderly people and their relationship with the pandemic. What can pose security risks?

Response: We have made the link between staff and residents’ safety with the pandemic clearer. The text now reads as follow:

The effects of the pandemic in nursing homes worldwide have taken a toll on all of us and in our humanity, as these facilities have been affected in an unprecedented manner [5]. The initial responses to the coronavirus disease (COVID-19) in nursing homes were limited. This depicts a poor preparedness of nursing homes in managing a public health crisis, albeit past serious concerns of potential threats [7, 8], with negative effects on the safety of staff and residents, exposing them to a hazardous and potentially life-threatening environment on a daily basis. This concerning unpreparedness uncovered long-lasting caveats in nursing homes, such as insufficient coordination with the health care system, generalized underfunding, and insufficient and inadequate workforce in the sector [9, 10].

In the method including the type of reserarch with theoretical referencial. In the method, it is necessary to include the approval protocol of the Research Ethics Committee, considering that the research involved data from institutions and participants.

Response: In the Methods we clearly define the study design and population: “We conducted a cross-sectional survey-based study in nursing homes of 2 regions in Southern Portugal (Algarve and Alentejo), in the early phases of the COVID-19 pandemic (March to July 2020)”. We have also clarified the reasons whereby an ethical review and approval were waived for this observational study under the current Portuguese legislation. We also wish to stress that prior to analysis and reporting, all personal data were anonymized. The Ethical statement now reads as follows:

Ethical review and approval were waived for this study, due to its non-intrusive and non-interventional characteristics; the anonymization of all personal data prior to analysis and reporting; the existing protocol between the Social Security Institute and the ABC research institute; and the urgency of gathering key information about the problems that nursing homes had been facing in a timely fashion.

In the conclusions is necessary including the reach of objective The objective of this study was to assess nursing homes’ 22 COVID-19 preparedness in Southern Portugal, including personnel’s work experiences during the 23 pandemic. 

Response: We followed the Reviewer’s suggestion and restated the objectives of our study in the Conclusion section.

Reviewer 3 Report

This is a worthwhile study, although it is difficult to interpret. There are a few issues that need to be resolved before it can be published.

Please add a description of external validity. Is there any possibility of applying this research outside of Portugal?
Please keep your conclusion concise.

P3, Line 122 Please clarify who "personnel" is referring to.
P4, Line 149 Please clarify this sentence.  I could not catch the meaning.

P4, Line 179  Delete the instructions.

Translated with www.DeepL.com/Translator (free version)

Author Response

We extend our sincere thanks to the three Reviewers for their constructive suggestions. A point-by-point response to Reviewer 3 follows. All necessary changes have been incorporated into the revised version of the manuscript.

Kind regards,

The Authors

Point-by-point response to Reviewer 3

This is a worthwhile study, although it is difficult to interpret. There are a few issues that need to be resolved before it can be published.

Please add a description of external validity. Is there any possibility of applying this research outside of Portugal?

Response: We have revised the Study Strengths and Limitations sub-section to address the generalizability of our findings outside the context of the study.

P3, Line 122 Please clarify who "personnel" is referring to.

Response: Following the suggestion of another Reviewer, we dropped the use of the term personnel when referring to nursing home staff.

P4, Line 149 Please clarify this sentence.  I could not catch the meaning.

Response: In our survey, the last question was open-ended, and we asked nursing home staff if there were any other aspects of their experience with dealing with the pandemic, they would like to share with the research team. The sentence now reads as follows:

The last question in the survey was an open-ended question where respondents could share, in their own words, how they have been experiencing the pandemic crisis.

P4, Line 179  Delete the instructions.

Response: We thank the Reviewer for flagging this issue. We have deleted the journal’s template instructions from the manuscript.

Round 2

Reviewer 2 Report

Dear autors,

Good work. I send my impressions for the Editor.

Best Regards,

The Evaluator